# COT Flow: Learning Optimal-Transport Image Sampling and Editing by Contrastive Pairs

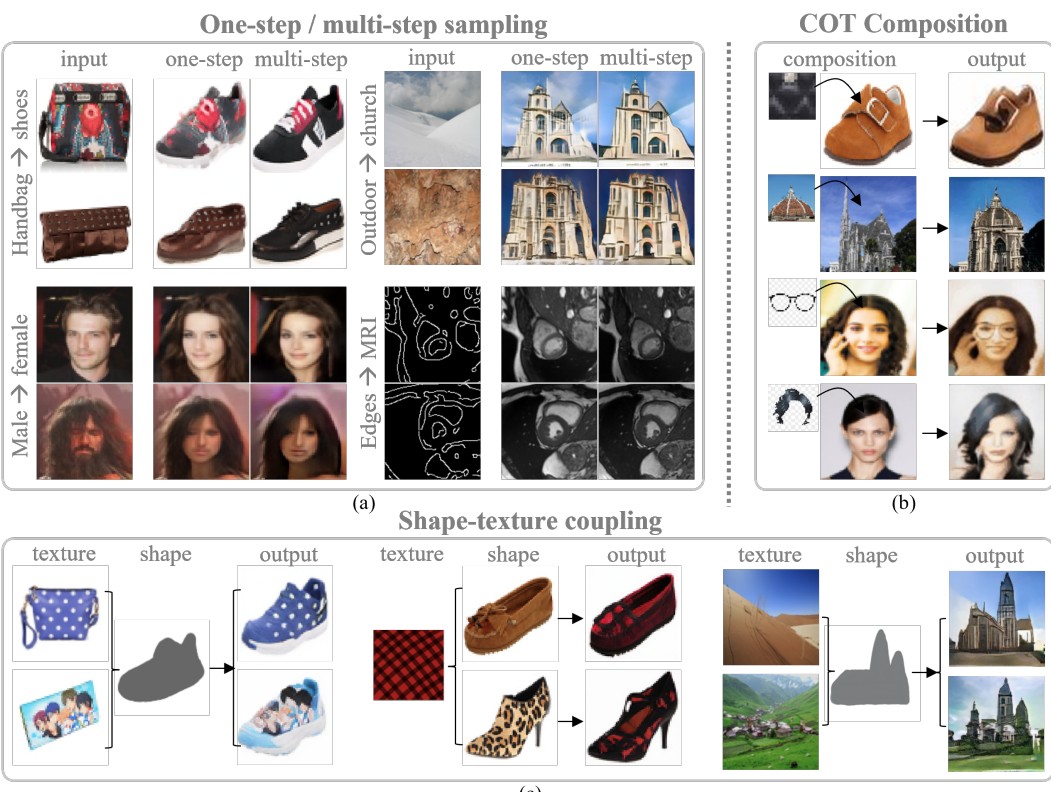

Figure 1: (a). Unpaired image-to-image translation by our proposed COT Flow, with one-step or multi-step sampling. (b), (c). Our proposed COT Editor enables zero-shot image editing with high flexibility. COT composition (b) allows users to composite elements and synthesize realistic images. Shape-texture coupling (c) allows users to separately draw or use shapes and textures as dual inputs, to generate fused images with high quality.

## Abstract

Diffusion models have demonstrated strong performance in sampling and editing multi-modal data with high generation quality, yet they suffer from the iterative generation process which is computationally expensive and slow. In addition, most methods are constrained to generate data from Gaussian noise, which limits their sampling and editing flexibility. To overcome both disadvantages, we present *Contrastive Optimal Transport Flow (COT Flow)*, a new method that achieves fast and high-quality generation with improved zero-shot editing flexibility compared to previous diffusion models. Benefiting from optimal transport (OT), our method has no limitation on the prior distribution, enabling unpaired image-to-image (I2I) translation and *doubling* the editable space (at both the start and end of the trajectory) compared to other zero-shot editing methods. In terms of quality, COT

Flow can generate competitive results in merely one step compared to previous state-of-the-art unpaired image-to-image (I2I) translation methods. To highlight the advantages of COT Flow through the introduction of OT, we introduce the *COT Editor* to perform user-guided editing with excellent flexibility and quality.

# 1 INTRODUCTION

Diffusion models, with flexible training and sampling principles rooted in Statistical Physics, have achieved unprecedented success in generating data from noise Ho et al. (2020); Song & Ermon (2019); Ramesh et al. (2022); Saharia et al. (2022); Nichol et al. (2021); Rombach et al. (2021); Dhariwal & Nichol (2021); Ho & Salimans (2022); Kazerouni et al. (2023); Janner et al. (2022); Poole et al. (2022); Li et al. (2023); Liu et al. (2024). However, the fundamental limitations of diffusion-based models, namely the sampling inefficiency and restrictive prior distribution, still barricade them from wider applications, despite the recent series of improved methods Nichol & Dhariwal (2021); Karras et al. (2022); Song et al. (2023). With a similar iterative sampling process, flow-based methods Chen et al. (2018); Kidger et al. (2020) also suffer from the computational inefficiency problem. From a high-level perspective, the current deep generative models still cannot simultaneously satisfy three performance indicators: (1) high-quality generation, (2) mode coverage and diversity, and (3) fast sampling, which is identified as the *generative learning trilemma* Xiao et al. (2021) shown in Fig.2a.

To tackle the generative learning trilemma and eliminate the constraints on prior distribution, we present a novel flow-based model called *Contrastive Optimal Transport Flow (COT Flow)*, which fundamentally addresses the computational inefficiency problem through the optimal transport (OT) formulation. We claim that OT enables the *fastest* sampling for diffusion/flow-based methods with two key features to overcome sampling inefficiency: (1) straight lines from source to target and (2) no crossing among the trajectories. Similar principles were approached implicitly in a few latest work Liu et al. (2022); Lipman et al. (2022); Tong et al. (2023); Esser et al. (2024); Karras et al. (2022). Specifically, many recent breakthroughs Song et al. (2020a); Nichol & Dhariwal (2021); Karras et al. (2022) focused on the following strategies: optimizing the sampling trajectories towards straight lines, improving the time schedule of the diffusion process Song et al. (2020a); Karras et al. (2022), adjusting the noise schedule or forward diffusion process Song et al. (2020b); Nichol & Dhariwal (2021); Lin et al. (2023); Bartosh et al. (2024), introducing fast samplers Nichol & Dhariwal (2021); Lu et al. (2022a;b); Karras et al. (2022), using distillation techniques Song et al. (2023); Liu et al. (2022); Salimans & Ho (2022); Xu et al. (2023); Meng et al. (2022), and eliminating the crossing among the trajectories to improve sample stability and efficiencyLiu et al. (2022); Lipman et al. (2022); Tong et al. (2023); Esser et al. (2024). We note that these improved techniques, though from different angles, approached the similar concept of OT between Gaussian and data distribution, as shown in Fig.2b. Another prominent group of recent works (Korotin et al. (2022a;b); Fan et al. (2021; 2022); Rout et al. (2021)) enforce direct OT by training two neural networks on saddle point problems Boyd & Vandenberghe (2004).

The proposed COT Flow satisfies the three performance requirements in the trilemma:

**Sample efficiency:** The proposed COT Flow explicitly builds the bridge between diffusion/flow-based models and OT, and thus enforces straight trajectories and eliminates the crossing to improve sample efficiency. With the benefit of both diffusion/flow-based models and the OT formulation, COT Flow enables one-step or few-step sampling by design, while still producing high-quality and high-diversity results from arbitrary prior distributions. Furthermore, COT Flow allows zero-shot editing, and introduces diverse editing possibilities (Fig.1b).

**Sample quality:** COT Flow leverages the intriguing similarities between consistency models Song et al. (2023); Song & Dhariwal (2023); Luo et al. (2023) and contrastive learning He et al. (2019); Chen & He (2020); Chen et al. (2020); Grill et al. (2020) to produce high-quality generation using indirect loss functions. In particular, the objective of consistency models consists of the similarity between time-adjacent data pairs $\langle \mathbf{x}_t, \mathbf{x}_{t+1} \rangle$, which function exactly the same as the positive sample pairs in contrastive learning (He et al. (2019) Eq.1). In addition, consistency models use a series of similar techniques as those in contrastive learning, such as exponential moving average (EMA) weights of the teacher model and "stopgrad" operator Song et al. (2023); Song & Dhariwal (2023); Chen & He (2020); Grill et al. (2020), suggesting the hidden link between the two state-of-the-art

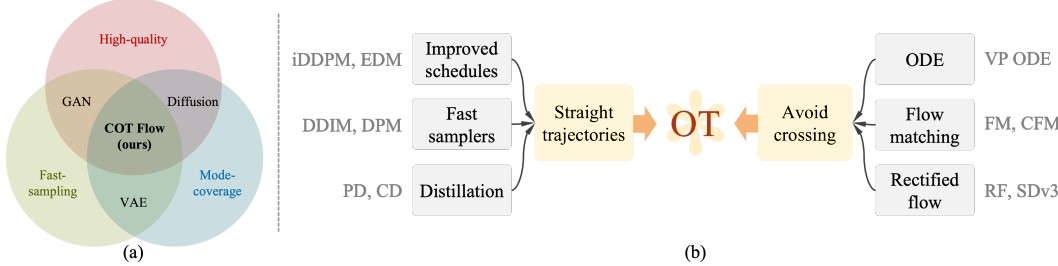

Figure 2: (a). The generative learning trilemma. Current generative methods still cannot simultaneously satisfy the three performance indicators: high quality, fast sampling, and mode coverage. (b). Recent developments of the diffusion/flow-based generative models, including iDDPMNichol & Dhariwal (2021), EDMKarras et al. (2022), DDIMSong et al. (2020a), DPMLu et al. (2022a), Progressive Distillation (PD)Salimans & Ho (2022), Consistency Distillation (CD)Song et al. (2023), VP ODESong et al. (2020b), Flow Matching (FM)Lipman et al. (2022), Conditional Flow Matching (CFM)Tong et al. (2023), Rectified Flow (RF)Liu et al. (2022), Stable Diffusion v3 (SDv3)Esser et al. (2024) All methods implicitly approach the OT formulation, either by sampling straight trajectories or avoiding crossing between the trajectories through various techniques.)

learning frameworks. Enlightened by this connection, we introduce the *Contrastive OT Pairs (COT Pairs)* for positive pair sampling during COT Flow training. By using a similar contrastive loss as in Grill et al. (2020), we consider the proposed COT Flow model as a powerful contrastive learning encoder $\mathcal{E}$ to map all data points on the OT trajectories towards their end. We evaluate COT Flow's sample quality via the FID scores in various unpaired I2I translation tasks such as handbags→shoes, CelebA male→female, and outdoor→church (Fig.1a).

**Mode coverage:** COT Flow achieves competitive sample diversity and mode coverage compared to diffusion models, benefiting from the non-adversarial contrastive loss and the OT formulation. The adversarial objectives in Generative Adversarial Nets (GAN)Goodfellow et al. (2014), Wasserstein GANArjovsky et al. (2017), and StyleGANKarras et al. (2018; 2019) are susceptible to training instability and mode collapse Xiao et al. (2021), which even the state-of-the-art GAN-based methods still suffer from Park et al. (2020). Diffusion-based model objectives, on the other hand, are closely related to the Evidence Lower Bound (ELBO) of the target data and are thus less prone to training instability and mode collapse Ho et al. (2020); Kingma & Gao (2023). In addition, with the OT formulation, the proposed COT Flow minimizes the transportation cost and directly maps the source distribution to the target distribution, improving the faithfulness to the target data.

In summary, our main contributions are: **(1)** We tackle the generative learning trilemma by introducing a novel framework called Contrastive Optimal Transport Flow (COT Flow), which explicitly combines diffusion/flow-based model with OT to directly learn the generative flow between any two unpaired data sources. **(2)** We present the Contrastive Optimal Transport Pair (COT Pair) formulation to train our proposed COT Flow, leveraging the intriguing connection between consistency models and contrastive learning. **(3)** To showcase the advantages of COT Flow, we introduce the *COT Editor* to perform controllable sampling and flexible zero-shot image editing, including COT composition, shape-texture coupling, and COT augmentation, and demonstrate these functionalities via diverse data and application scenarios.

## 2 BACKGROUND

COT Flow leverages the theories and concepts from (1) optimal transport Villani (2009), (2) contrastive learning He et al. (2019), and (3) consistency models Song et al. (2023), crossing these three prominent methodologies in optimization and machine learning. For a quick understanding of the proposed COT Flow, we first briefly present the three core methodologies and discuss their interconnections in Section 3.1.

**Notations.** Throughout the paper, $\mathcal{X}$ and $\mathcal{Y}$ denote two metric spaces of data, $\mu(\mathbf{x})$ and $\nu(\mathbf{y})$ denote the probability distributions on $\mathcal{X}$ and $\mathcal{Y}$, respectively. For describing the projection between $\mu(\mathbf{x})$

and $\nu(\mathbf{y})$, we denote $T : \mathcal{X} \to \mathcal{Y}$ as a measurable map, which satisfies: for any measurable subsets $B \subset \mathcal{Y}$, $T^{-1}(B) \subset \mathcal{X}$. We denote $\Pi(\mu, \nu)$ as the set of joint probability distributions on $\mathcal{X} \times \mathcal{Y}$ whose marginals are $\mu$ and $\nu$.

## 2.1 Optimal Transport

The optimal Transport (OT) problem seeks the minimum overall transportation cost from one measure to another. Consider a cost function $c : \mathcal{X} \times \mathcal{Y} \to \mathbb{R}$, Kantorovitch (1958) formulates a transport coupling $\pi \in \Pi(\mu, \nu)$ and introduces the OT cost:

$$\text{Cost}(\mu, \nu) := \inf_{\pi \in \Pi(\mu, \nu)} \int_{\mathcal{X} \times \mathcal{Y}} c(\mathbf{x}, \mathbf{y}) d\pi(\mathbf{x}, \mathbf{y}) \tag{1}$$

This is defined as the Kantorovich problem, where the infimum is taken over transport couplings $\pi \in \Pi(\mu, \nu)$. The optimal $\pi^*$ is called the OT plan, which always exists under mild conditions on spaces $\mathcal{X}$, $\mathcal{Y}$ and cost function $c$ (Villani (2009)). According to the duality principle Boyd & Vandenberghe (2004), the dual problem of Kantorovich's optimization is:

$$\text{Cost}(\mu, \nu) := \sup_{\varphi, \psi} \left\{ \int_{\mathcal{X}} \varphi(\mathbf{x}) d\mu(\mathbf{x}) + \int_{\mathcal{Y}} \psi(\mathbf{y}) d\nu(\mathbf{y}) \right\} \tag{2}$$

where $\varphi \in L^1(\mu)$ and $\psi \in L^1(\nu)$ are called Kantorovich potentials which satisfy $\varphi(\mathbf{x}) + \psi(\mathbf{y}) \leq c(\mathbf{x}, \mathbf{y})$. For $\varphi : \mathcal{X} \to \mathbb{R}$, $\psi : \mathcal{Y} \to \mathbb{R}$, and a certain cost function $c$, we replace the first potential $\varphi(\mathbf{x})$ by defining the $c$-transform of $\psi$: $\varphi(\mathbf{x}) = \psi^c(\mathbf{x}) = \inf_{\mathbf{y} \in \mathcal{Y}} \{c(\mathbf{x}, \mathbf{y}) - \psi(\mathbf{y})\}$, and the Kantorovich problem 2 is rewritten as:

$$\text{Cost}(\mu, \nu) := \sup_{\psi} \left\{ \int_{\mathcal{X}} \inf_{\mathbf{y}} \{c(\mathbf{x}, \mathbf{y}) - \psi(\mathbf{y})\} d\mu(\mathbf{x}) + \int_{\mathcal{Y}} \psi(\mathbf{y}) d\nu(\mathbf{y}) \right\} \tag{3}$$

where we denote the right side of 3 as a saddle point problem $\sup_{\psi} \inf_{\mathbf{y}} \mathcal{L}(\psi, \mathbf{y})$, whose solution $(\psi^*, \mathbf{y}^*)$ contains the optimal choice of $\mathbf{y}$ given a certain $\mathbf{x}$. In practice, $\mathbf{y}^*$ can be estimated by optimizing a neural network $\tilde{\mathbf{y}} = T_\theta(\mathbf{x})$, leading to neural OT methods Korotin et al. (2022a;b); Fan et al. (2021; 2022). We further illustrate the training of $T_\theta(\mathbf{x})$ in Section 3.

## 2.2 Contrastive Learning

With impressive results on multiple visual tasks, contrastive learning methods learn data representations by attracting the embeddings of positive sample pairs and (optionally) repulse the embeddings of negative sample pairs in an unsupervised manner He et al. (2019); Chen et al. (2020). For the methods that only consider the positive pairs Chen & He (2020); Grill et al. (2020), the core methodology can be described as minimizing the loss function:

$$\mathcal{L}(\theta, \theta^-) := d(q_\theta(\mathcal{E}_\theta(\mathbf{x})), \mathcal{E}_{\theta^-}(\mathbf{x}^+)) \tag{4}$$

where $\mathcal{E}$ is the target network, which we consider as an encoder. $\theta^-$ denotes the exponential moving average (EMA) of the past values of the network's weights $\theta$. $d(\cdot, \cdot)$ is the distance function between the data embedding $\mathcal{E}(\mathbf{x})$ and its corresponding positive pairs $\mathcal{E}(\mathbf{x}^+)$, whose inputs $\mathbf{x}^+$ are augmented from the same sample $\mathbf{x}$. Combined with the EMA weights $\theta^-$ and the "stopgrad" operator, an additional prediction head $q_\theta$ is introduced on top of the encoder $\mathcal{E}_\theta$ to prevent model collapse and enable the contrastive learning methods to produce meaningful representations. In Section 3, we introduce the similarities between contrastive learning and consistency models.

## 2.3 Consistency Models

Consistency models (CMs) are an emerging family of generative models whose key idea is maintaining consistency along the ordinary differential equation (ODE) trajectory derived from the diffusion models, which we briefly introduce in Appendix E. One drawback of diffusion models is their slow sampling speed. CMs, on the other hand, learn the consistency along the trajectories $\{\hat{\mathbf{x}}_t\}_{t \in [0,T]}$ of the probability flow ODE 28 and map all the points on these trajectories to their origin $\hat{\mathbf{x}}_0$. This mapping can be described as the consistency function $\mathbf{f}^* : (\mathbf{x}_t, t) \to \mathbf{x}_0$ which satisfies the boundary condition $\mathbf{f}^*(\mathbf{x}, 0) = \mathbf{x}_0$. We then approximate $\mathbf{f}^*(\mathbf{x}, t)$ by training the consistency model $\mathbf{f}_\theta(\mathbf{x}_t, t)$.

By discretizing the probability flow ODE 28 with a limited sequence of time steps $\epsilon < t_1 < t_2 < ... < t_N = T$, the consistency model $\mathbf{f}_\theta(\mathbf{x}_t, t)$ is trained by minimizing the consistency matching loss (CM loss):

$$\mathcal{L}^N(\theta, \theta^-) := \mathbb{E}\big[\lambda(t_i)d(\mathbf{f}_\theta(\mathbf{x}_{t_{i+1}}, t_{i+1}), \mathbf{f}_{\theta^-}(\mathbf{x}_{t_i}, t_i))\big], i \sim \mathcal{U}[1, N-1] \quad (5)$$

where $\mathbf{x}_{t_{i+1}}$ is sampled from the distribution $p_{t_{i+1}}(\mathbf{x})$ and the parameter $\theta^-$ is the EMA of $\theta$ obtained with the "stopgrad" operator $\theta^- \leftarrow \text{stopgrad}(\mu\theta^- + (1-\mu)\theta)$. $0 \leq \mu < 1$ denotes the EMA decay rate. $\lambda(t_i) > 0$ is a weighting function and $d(\cdot, \cdot)$ is a distance function with a typical choice of squared $l_2$. $\mathcal{U}[1, N-1]$ denotes the uniform distribution over $1, 2, ..., N-1$. For $\mathbf{x}_{t_i}$, CMs provide two approximations and correspondingly form two training algorithms called consistency distillation (CD) and consistency training (CT). The approximation from CD is $\hat{\mathbf{x}}_{t_i} = \mathbf{x}_{t_{i+1}} - (t_i - t_{i+1})t_{i+1}\mathbf{s}_\phi(\mathbf{x}_{t_{i+1}}, t_{i+1})$, which relies on a pre-trained diffusion model $\mathbf{s}_\phi(\mathbf{x}, t)$. While the approximation from CT is $\hat{\mathbf{x}}_{t_i} = \mathbf{x} + t_i\mathbf{z}$ where $\mathbf{z} \sim \mathcal{N}(\mathbf{0}, \mathbf{I})$ is the same noise when forming $\mathbf{x}_{t_{i+1}} = \mathbf{x} + t_{i+1}\mathbf{z}$. We can directly sample the final generation by $\mathbf{x}_0 = \mathbf{f}_\theta(\mathbf{z}, t_N)$ or optionally sample the intermediate results $\mathbf{x}_k = \mathbf{f}_\theta(\mathbf{x}_{k+1}, t_{i_{k+1}}) + \sqrt{t_N^2 - \epsilon^2}\mathbf{z}_k$ for $k = K-1, ..., 1$.

Comparing the CM loss $\mathcal{L}^N(\theta, \theta^-)$ in 5 and the contrastive learning loss in 4, we observe both structural and conceptual similarities between them, which will be discussed in Section 3.1.

## 3 METHOD

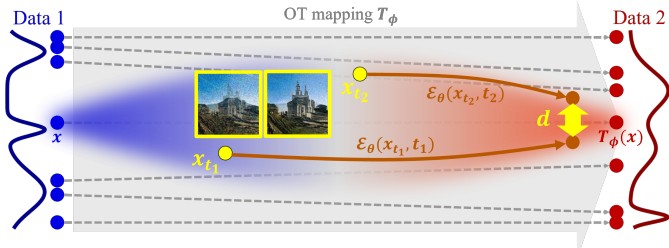

Figure 3: An overview of the training process. COT Flow minimizes the distances between the encodings of the positive pairs, which are sampled in the augmentation area between **x** in Data 1 and its OT mapping $T_\phi(\mathbf{x})$ (Eq.12).

Our proposed COT Flow tackles the generative learning trilemma by fundamentally regularizing the transportation flows between two distributions. COT Flow consists of three main parts: (1) COT Pairs, (2) COT training, and (3) COT Editor. In the sections below, we first discuss the similarities between CMs and contrastive learning, which inspire our formulation of COT Pairs and COT training. We then introduce the COT Editor framework.

### 3.1 SIMILARITIES BETWEEN CONTRASTIVE LEARNING AND CONSISTENCY MODELS

One may raise a question on the mechanism of CMs: Why do they work well by simply minimizing the difference between two points on the same trajectory, especially with no guidance of the trajectory's origin $\mathbf{x}_0$ in the loss function? Here we put forward a hypothesis on why they learn to map to the origin by exploring the systematic similarities between CMs and contrastive learning: *The consistency function $\mathbf{f}_\theta(\mathbf{x}, t)$ is a trajectory's origin encoder $\mathcal{E}_\theta(\mathbf{x})$, which has the same functionality of the encoder in contrastive learning.*

Firstly, we notice the similarity between the CM loss 5 and the contrastive loss 4, which are both summarized by a distance metric $d(\cdot, \cdot)$. Specifically, the CM loss indicates the distance between the two output points $\mathbf{f}_\theta(\mathbf{x}_{t_{i+1}}, t_{i+1})$ and $\mathbf{f}_{\theta^-}(\mathbf{x}_{t_i}, t_i)$ from the same trajectory, while the contrastive loss indicates the distance between the embeddings of the positive pairs $\mathcal{E}_\theta(\mathbf{x})$ and $\mathcal{E}_{\theta^-}(\mathbf{x}^+)$ from the same image. This suggests that CMs have the capability of learning representations from complex distributions and are capable of mapping denoising trajectories $\{\mathbf{x}_t\}_{t \in [\epsilon, T]}$ to their origins $\mathbf{x}_0$.

Secondly, the strategies and training recipes of the two methods are similar, especially those for preventing mode collapsing. They both utilize weight-sharing Siamese networks $\theta, \theta^-$ to minimize the distance metric $d(\cdot, \cdot)$ of the entities, and they both use "stopgrad" operations to distinguish the networks and prevent collapsing:

$$\theta^- \leftarrow \theta^- - \eta\nabla_\theta d\big(\mathcal{E}_\theta(\cdot), \text{stopgrad}(\mathcal{E}_{\theta^-}(\cdot))\big) \quad (6)$$

Furthermore, the recent work from both sides Chen & He (2020); Song & Dhariwal (2023) illustrated a common improvement to optimize the results and simplify the strategies: removing the EMA decay for the Siamese structure, whose weights share the same update $\nabla_\theta$. This improvement has been proven effective from both sides Chen & He (2020); Song & Dhariwal (2023), underlining the same mechanism between CMs and contrastive learning.

With the above observations, we explain the capability of the consistency function $\mathbf{f}_\theta(\mathbf{x}, t)$ to map the intermediates towards the origin by considering the consistency function $\mathbf{f}_\theta(\mathbf{x}, t)$ as the encoder $\mathcal{E}_\theta(\mathbf{x})$ in contrastive learning. With this foundation, we introduce COT Pairs and COT training in the following sections.

### 3.2 COT PAIRS

In Section 2.1, we introduce the Kantorivich problem. The entropic regularization of the Kantorovich problem, namely the entropic OT (EOT) problem Villani (2009), minimizes the transportation cost derived from 1:

$$\text{Cost}(\mu, \nu) := \inf_{\pi \in \Pi(\mu, \nu)} \left\{ \int_{\mathcal{X} \times \mathcal{Y}} c(\mathbf{x}, \mathbf{y}) d\pi(\mathbf{x}, \mathbf{y}) + \lambda H(\pi) \right\} \tag{7}$$

where the solution $\pi_\lambda^*$ is the EOT plan. With the relative entropy $\lambda H(\pi)$, the expensive computation in the exact OT problem is alleviated. For neural OT models, using EOT enables stochastic processes within the OT mapping and relates OT with diffusion models Gushchin et al. (2022). In the following Eq.12, we introduce noise into COT training, where Proposition 3.1 shows its relationship to the EOT plan.

We modify a neural OT model to estimate the OT map between the two data distributions. According to Section 2.1, the solution $(\psi^*, \mathbf{y}^*)$ of the Kantorovich problem 3 can be estimated by two corresponding networks $(\psi_\omega, T_\phi(\mathbf{x}))$, resulting in the neural OT objective:

$$\text{Cost}(\mu, \nu) := \sup_{\psi_\omega} \left\{ \inf_{T_\phi} \int_{\mathcal{X}} \left\{ c(\mathbf{x}, T_\phi(\mathbf{x})) - \psi_\omega(T_\phi(\mathbf{x})) \right\} d\mu(\mathbf{x}) + \int_{\mathcal{Y}} \psi_\omega(\mathbf{y}) d\nu(\mathbf{y}) \right\} \tag{8}$$

where $\psi_\omega$ denotes the Kantorovich potential in Section 2.1 and $T_\phi$ is the estimated OT map. The infimum of $T_\phi$ is interchanged with the integral by Rockafellar (1976) and the OT problem 1 is derived into the optimization of the neural networks:

$$\sup_\omega \inf_\phi \mathcal{L}(\psi_\omega, T_\phi) \tag{9}$$

To approach 9 in implementation, we optimize the parameters $\omega$, $\phi$ using stochastic gradient ascent-descent (SGAD) by sampling mini-batch data from source and target datasets $\mathbf{x} \sim \mu(\mathbf{x})$, $\mathbf{y} \sim \nu(\mathbf{y})$:

$$\omega \leftarrow \omega + \nabla_\omega \left\{ -\frac{1}{|\mathbf{x}|} \sum_{\mathbf{x} \in \mathcal{X}} \psi_\omega(T_\phi(\mathbf{x})) + \frac{1}{|\mathbf{y}|} \sum_{\mathbf{y} \in \mathcal{Y}} \psi_\omega(\mathbf{y}) \right\} \tag{10}$$

$$\phi \leftarrow \phi - \nabla_\phi \left\{ \frac{1}{|\mathbf{x}|} \sum_{\mathbf{x} \in \mathcal{X}} \left[ c(\mathbf{x}, T_\phi(\mathbf{x})) - \psi_\omega(T_\phi(\mathbf{x})) \right] \right\} \tag{11}$$

where $|\mathbf{x}|$, $|\mathbf{y}|$ denote the sizes of the corresponding mini-batches $\mathbf{x} \sim \nu(\mathbf{x})$, $\mathbf{y} \sim \mu(\mathbf{y})$. $c(\cdot, \cdot)$ denotes the cost function in 3 which is typically $l_2$-norm. Based on the trained $T_\phi(\mathbf{x})$ in 10 and 11, we interpolate an augmentation area between $\mu(\mathbf{x})$ and $\nu(\mathbf{y})$ for training COT Flow, whose concept "augmentation" derives from contrastive learning:

$$\{\tilde{\mathbf{x}}_t\}_{t \in [0,1]} = \{tT_\phi(\mathbf{x}) + (1 - t)\mathbf{x} + t(1 - t)\sigma^2 \mathbf{z}\}_{t \in [0,1]} \tag{12}$$

where $\sigma$ is the noise scale and $\mathbf{z} \sim \mathcal{N}(\mathbf{0}, \mathbf{I})$ is standard Gaussian noise. We prove that the OT plan $\pi^*$ in 1 can be extended in $t \in [0, 1]$ by formulating this augmentation area:

**Proposition 3.1** (Eq.12 estimates the dynamic extension of the OT plan). *Let $\pi^*$ be the OT plan between $\mu(\mathbf{x})$ and $\nu(\mathbf{y})$. Let the OT map $T^*$ recovers $\pi^*$. The augmentation defined by Eq.12 using $T^*$ samples the same probability as the dynamic extension of the EOT plan $\pi_\lambda^*$ with $\lambda = 2\sigma^2$.*

We provide the proof in Appendix B. With the guarantee of Proposition 3.1 and the observation in Section 3.1, we consider the augmentations $\tilde{\mathbf{x}}_t$ as the intermediates of the entropic OT trajectory $\{\tilde{\mathbf{x}}_t\}_{t\in[0,1]}$ and formulate a set of positive pairs as in contrastive learning, which we name as COT Pairs. In particular, COT Pairs $\langle \mathbf{x}_{t_1}, \mathbf{x}_{t_2} \rangle$ are randomly selected along the trajectory $\{\tilde{\mathbf{x}}_t\}_{t\in[0,1]}$:

$$\mathbf{x}_{t_1}, \mathbf{x}_{t_2} \in \{\tilde{\mathbf{x}}_t\}_{t\in[0,1]}, \quad 0 \le t_1 < t_2 \le 1 \tag{13}$$

Unlike CMs choosing adjacent pairs from ODE solvers, we formulate random COT pairs in the proposed augmentation area in Eq.12.

## 3.3 COT TRAINING

According to the relationship between contrastive learning and CMs discussed in section 3.1, we consider the consistency function $\mathbf{f}_\theta(\cdot)$ as an encoder $\mathcal{E}(\mathbf{x}_t)$ towards the origins $\mathbf{y}$ of the entropic OT trajectories $\{\tilde{\mathbf{x}}_t\}_{t\in[0,1]}$. The COT training loss to optimize the origin encoder $\mathcal{E}(\mathbf{x}_t, t)$ is:

$$\mathcal{L}_{\text{COT}}(\theta) = d\big(\mathcal{E}_\theta(\mathbf{x}_{t_1}, t_1), \mathcal{E}_\theta(\mathbf{x}_{t_2}, t_2)\big), \quad 0 \le t_1 < t_2 \le 1 \tag{14}$$

where $d(\cdot, \cdot)$ denotes the dissimilarity function, which is $l_2$-norm by default and $\mathbf{x}_{t_1}, \mathbf{x}_{t_2}$ is the COT Pair from $\{\tilde{\mathbf{x}}_t\}_{t\in[0,1]}$. Inspired by Esser et al. (2024), the origin estimation $\mathcal{E}(\mathbf{x}_t)$ is more difficult for $t$ in the middle of $[0,1]$ since we introduce additional Gaussian noise in a quadratic manner $t(1-t)\sigma^2\mathbf{z}$. We use the mode distribution defined in Esser et al. (2024) to sample the intermediate time step with higher frequencies.

Compared to the CM loss in 4, we emphasize the consistency along the whole OT trajectory through COT Pairs in random time steps. In addition, we use auxiliary noise to enhance the robustness of the OT consistency, with the theoretical guarantee in EOT and Lemma 3.1. The pseudo-code of COT Flow training pipeline is in Algorithm 1. The detailed algorithm in implementation is in Appendix A.

---

**Algorithm 1** COT Training

---

**Input:** source data distribution $\mu$, neural OT map $T_\phi$, parameters $\theta$, noise scale $\sigma$, learning rate $\eta$.
**repeat**
    Sample $\mathbf{x} \sim \mu(\mathbf{x})$ and $t_1, t_2 \in [0,1]$
    $\tilde{\mathbf{x}}_{t_i} \leftarrow t_i T_\phi(\mathbf{x}) + (1-t_i)\mathbf{x} + t_i(1-t_i)\sigma^2\mathbf{z}, \quad \mathbf{z} \sim \mathcal{N}(\mathbf{0}, \mathbf{I}), \quad i = 1, 2$
    $\mathcal{L}_{\text{COT}}(\theta) \leftarrow d\big(\mathcal{E}_\theta(\mathbf{x}_{t_1}, t_1), \mathcal{E}_\theta(\mathbf{x}_{t_2}, t_2)\big)$
    $\theta \leftarrow \text{stopgrad}(\theta + \eta\nabla_\theta\mathcal{L}_{\text{COT}}(\theta))$
**until** convergence

---

## 3.4 COT EDITOR

To further illustrate the flexibility and generalizability of COT Flow, we introduce COT Editor, a zero-shot image editor that possesses various scenarios using a series of modifications of a self-augmentation sampling strategy:

$$\tilde{\mathbf{x}}_{t_k}^{(k)} = t_k\mathbf{x} + (1-t_k)\tilde{\mathbf{y}}^{(k)} + t_k(1-t_k)\sigma^2\mathbf{z}_k \tag{15}$$

$$\tilde{\mathbf{y}}^{(k+1)} = \mathcal{E}_\theta\big(\tilde{\mathbf{x}}_{t_k}^{(k)}, t_k\big), \qquad k = 1, 2, \dots \tag{16}$$

where $\tilde{\mathbf{y}}^{(k)}$ is the last estimation of target data. $\tilde{\mathbf{x}}_{t_k}^{(k)}$ is the corresponding self-augmented sample. $t_k$ represents a chosen time step series $0 < t_k < 1$, which is not limited to monotonically increase over time. With a well-trained model under Eq.9, we can sample from the source distribution through one-step sampling $\tilde{\mathbf{y}} = \mathcal{E}_\theta(\mathbf{x}, 0)$, or optionally adopt a multi-step self-augmentation sampling strategy in Eq.15/21, which enables zero-shot editing through the intermediate sampling steps. Both sampling strategies are illustrated in the left panel of Fig.4. With the benefit of unlimited input distribution of COT Flow, COT Editor extends the existing zero-shot image editing scenarios, formulating a dual-channel editing space where both source and target data space $\mathcal{X}, \mathcal{Y}$ are included. We demonstrate its capability by introducing the following scenarios: (1) COT composition, (2) shape-texture coupling, and (3) COT augmentation.

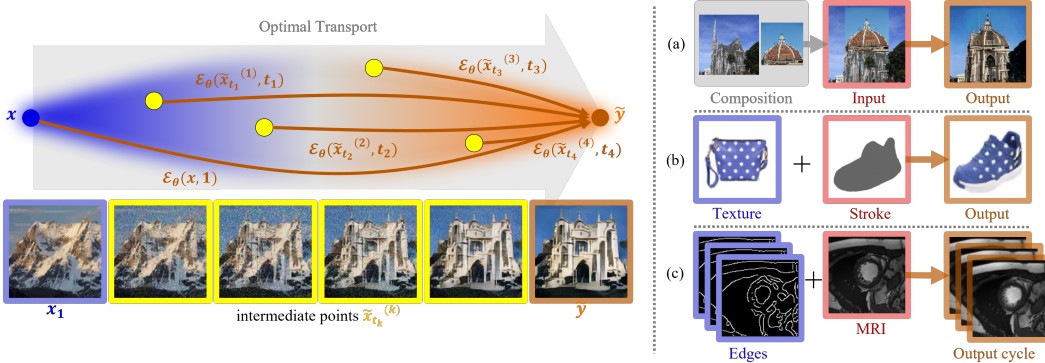

Figure 4: **Left:** The sampling strategy of our method. Given an input $\mathbf{x}$, we can generate the target data $\tilde{\mathbf{y}}$ with one-step sampling $\tilde{\mathbf{y}} = \mathcal{E}_\theta(\mathbf{x}, 1)$, or optionally multi-step sampling using Eq.15/21, where the intermediates $\tilde{\mathbf{x}}_{t_k}$ are the augmentations between the source input $\mathbf{x}$ and the generated target $\tilde{\mathbf{y}}$. **Right:** Three scenarios of the proposed COT Editor, some of which have dual-channel inputs as extensions to the current editing methods. (a). COT composition. Given a target image $\mathbf{y}$ with an edited component or mask $\mathbf{m}$, we use the guidance $\mathbf{y}^{(g)} = \mathbf{y} \oplus \mathbf{m}$ as the single input and synthesize the output $\tilde{\mathbf{y}}$ by Eq.17. (b). Shape-texture coupling. With a drawn stroke image $\hat{\mathbf{x}}_1$ and a texture image $\hat{\mathbf{x}}_2$, the output $\tilde{\mathbf{y}}$ consists of both features. (c). COT augmentation. Given a series of auto-detected cardiac-cycle edges $\{\hat{\mathbf{x}}^{(a)}\}$ and a single MRI $\mathbf{y}$, we can generate a cycle of cardiac MRI $\{\tilde{\mathbf{y}}\}$ with the same movements of $\{\hat{\mathbf{x}}^{(a)}\}$ and style of $\mathbf{y}$.

For COT composition, given a target image $\mathbf{y}$ with an edited component or mask $\mathbf{m}$, we denote the combination as the guidance $\mathbf{y}^{(g)} = \mathbf{y} \oplus \mathbf{m}$ of the COT Editor and perform the following one-step editing to obtain realistic outputs:

$$\tilde{\mathbf{y}} = \mathcal{E}_\theta(\mathbf{y}^{(g)} + t_g(1 - t_g)\sigma^2\mathbf{z}, t_g), \quad t_g \in [0, 1] \tag{17}$$

where $t_g$ denotes the chosen time step of the guidance editing, enabling the trade-off between faithfulness and realism as in Meng et al. (2021). For shape-texture coupling, considering a drawn shape $\hat{\mathbf{x}}_1$ and a texture image $\hat{\mathbf{x}}_2$, we can generate a realistic image using $\hat{\mathbf{x}}_1, \hat{\mathbf{x}}_2$ as the augmentation sources:

$$\tilde{\mathbf{y}} = \mathcal{E}_\theta(t_c\hat{\mathbf{x}}_1 + (1 - t_c)\hat{\mathbf{x}}_2 + t_c(1 - t_c)\sigma^2\mathbf{z}, t_c), \quad t_c \in [0, 1] \tag{18}$$

For COT augmentation, we provide a medical image synthesis scenario. We denote $\{\hat{\mathbf{x}}^{(a)}\}$ as a series of auto-detected cardiac-cycle edges and augment a fixed input cardiac MRI (cMRI) $\mathbf{y}$ by fusing them:

$$\{\tilde{\mathbf{y}}\} \leftarrow \mathcal{E}_\theta(t_a\mathbf{y} + (1 - t_a)\{\hat{\mathbf{x}}^{(a)}\} + t_a(1 - t_a)\sigma^2\mathbf{z}, t_a), \quad t_a \in [0, 1] \tag{19}$$

The dual ends of the OT trajectory in COT Flow enrich these additional zero-shot editing applications, where we demonstrate the results in Section 4.2.

## 4 EXPERIMENTS

We employ COT Flow in various experiments compared with other popular methods. Section 4.1 shows competitive performances of COT Flow on unpaired I2I translation benchmarks. We compare the generation quality with SDEdit Meng et al. (2021) and CycleGAN Zhu et al. (2017), which are popular diffusion/GAN-based methods. Section 4.2 provides the results of our proposed extended scenarios of zero-shot editing, including COT composition, shape-texture coupling, and COT augmentation. In Section 4.3, we discuss several key techniques of COT Flow by ablation studies. The implementation details of all the experiments are shown in Appendix A.

### 4.1 UNPAIRED IMAGE-TO-IMAGE TRANSLATION

We perform experiments on handbag→shoes (64×64), CelebA male→female (64×64), outdoor→church (128×128), and edges→cardiac MRI (cMRI) (128×128) to implement unpaired

Table 1: FID↓ scores of the baseline methods and our proposed COT Flow on handbag→shoes (64×64), CelebA male→female (64×64), and outdoor→church (128×128). Compared to SDEdit with a larger number of function evaluations (NFE), we use one-step sampling in COT Flow as the GAN-based methods.

| Method | DiscoGAN | CycleGAN | MUNIT | SDEdit | COT Flow (ours) |
|---|---|---|---|---|---|
| NFE | 1 | 1 | 1 | 500 | 1 |
| *handbag→shoes* | 22.42 | 16.00 | 15.76 | 18.91 | **15.01** |
| *male→female* | 35.64 | 17.74 | 17.07 | 17.26 | **16.30** |
| *outdoor→church* | 75.36 | 46.39 | 31.42 | 28.84 | **26.34** |

I2I translation. The formulation of these datasets is in Appendix A. With the recommendation of Karras et al. (2022) and Song et al. (2023) to train the diffusion-based methods, we choose the hyper-parameters that are unrelated to our proposed ideas to be in line with these methods, where further details can be found in Appendix A.

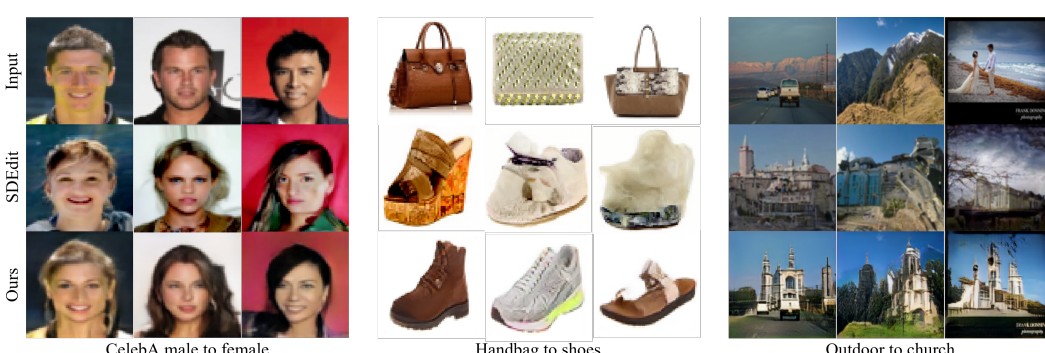

Figure 5: Generation comparison between our method (bottom row) and SDEdit (middle row) on CelebA male→female (64×64), handbag→shoes (64×64), and outdoor→church (128×128). We use one-step sampling in our method and set $t = 500$ of the reverse diffusion process in SDEdit to perform the results.

As shown in Fig.1a, our method provides high-quality generations with one-step or multi-step sampling. In Fig.5, we compare the generation results between SDEdit and the proposed COT Flow, illustrating a more faithful unpaired I2I translation by our method. In Table 1, our method outperforms the other diffusion/GAN-based methods in terms of the FID↓ scores by one-step sampling.

### 4.2 COT EDITOR SCENARIOS

In section3.4, we introduce three scenarios of the proposed COT Editor. Fig.1b further present editing results with the trained COT Flow on handbag→shoes (64×64), CelebA male→female (64×64), and outdoor→church (128×128).

### 4.3 ABLATION STUDIES

We provide reasons of COT Flow's key design by the following ablation studies. In Table 2, we choose alternated contrastive pair formulations, neural OT mapping direction, and sampling strategies, which represent the key design of our method. In particular, we (1) train a COT Flow model with only adjacent contrastive pairs $\langle \mathbf{x}_{t_k}, \mathbf{x}_{t_k+1} \rangle$ as is implemented in Song et al. (2023), (2) use the opposite direction of neural OT mapping from target to source ($T'(\mathbf{y})$) to form the contrastive pairs using $\{\tilde{\mathbf{x}}_t\}_{t \in [0,1]} = \{t\mathbf{y} + (1-t)T'_\phi(\mathbf{y}) + t(1-t)\sigma^2 \mathbf{z}\}_{t \in [0,1]}$ instead of Eq.12, and (3) try a different sampling strategy in an ancestral manner, which is commonly adopted in diffusion-based models Ho et al. (2020). As shown in Table 2, COT Flow with the paper's choice outperforms the other alternatives in one-step and multi-step sampling.

Table 2: Ablating COT pairs and sampling strategy on various datasets (evaluated by FID↓ scores). "Adjacent pairs" denotes training the COT Flow with only adjacent positive pairs $\langle \mathbf{x}_{t_k}, \mathbf{x}_{t_k+1} \rangle$ as is implemented in Song et al. (2023). "Reverse OT" denotes training a neural OT model $T'(\mathbf{y})$ with opposite direction mapping from target space $\mathcal{Y}$ to source space $\mathcal{X}$ and form the COT pairs. "Ancestral" denotes using a sampling strategy in an ancestral manner in COT Flow.

| Method | Adjacent pairs | Reverse NOT | Paper's choice | | |
|---|---|---|---|---|---|
| NFE | 1 | 1 | 40 (Ancestral) | 40 | 1 |
| *handbag→shoes* | 15.24 | 33.49 | 19.97 | 18.33 | **15.01** |
| *male→female* | 16.67 | 30.28 | 21.12 | 16.93 | **16.30** |
| *outdoor→church* | 26.95 | 38.11 | 26.92 | **26.05** | 26.34 |

## 5 CONCLUSION

We presented *COT Flow*, a new method that provides a tangible approach to tackle the generative learning trilemma, achieving fast and high-quality generation and flexible zero-shot image editing. Benefiting from OT reformulation, we achieved competitive sample quality on a great variety of unpaired I2I translation tasks, representing flow between diverse distributions. With the proposed *COT Editor*, We demonstrated flexible zero-shot editing capacities with three scenarios, namely, COT composition, shape-texture coupling, and COT augmentation.

Our method explicitly built the bridge between diffusion/flow-based models and OT by combining consistency models and contrastive learning, opening up new directions for future work. The proposed COT Editor expanded the possibility of zero-shot image editing by the dual-channel editing spaces, enabling new directions for zero-shot editing applications.

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

## A    IMPLEMENTATION DETAILS

In this section, we provide the implementation details of our method. Section A.1 provides the detailed training algorithm of our method in the implementation. Section A.2 introduces the used datasets and the construction of the unpaired I2I translation tasks. Section A.3 discusses the details of the chosen hyper-parameters of our method. Section A.4 provides the training details and the computational complexity of our method. Section A.5 introduces the alternative combinations of our method in Section 4.3 to perform the ablation studies.

## A.1 DETAILED ALGORITHM

In the implementation, we uniformly discretize the sampled time steps $t_1, t_2$ in Eq.13 with the number of the discrete time steps $N$. We use LPIPS Zhang et al. (2018) distance as the distance metric $d(\cdot, \cdot)$. The detailed training algorithm of our method is as follows:

---

**Algorithm 2** COT Training

---

**Input:** source data distribution $\mu$, neural OT map $T_\phi$, parameters $\theta$, noise scale $\sigma$, learning rate $\eta$, distance metric $d(\cdot, \cdot)$, and number of discretization $N$.

**repeat**

    Sample $\mathbf{x} \sim \mu(\mathbf{x})$ and $n_1, n_2 \in \mathcal{U}[0, N-1] \quad n_1 < n_2$

    $\tilde{\mathbf{x}}_{t_i} \leftarrow \frac{n_i}{N-1} T_\phi(\mathbf{x}) + (1 - \frac{n_i}{N-1})\mathbf{x} + \frac{n_i}{N-1}(1 - \frac{n_i}{N-1})\sigma^2 \mathbf{z}, \quad \mathbf{z} \sim \mathcal{N}(\mathbf{0}, \mathbf{I}), \quad i = 1, 2$

    $\theta_1, \theta_2 \leftarrow \theta$

    $\mathcal{L}_{\text{COT}}(\theta_1, \theta_2) \leftarrow d\big(\mathcal{E}_{\theta_1}(\mathbf{x}_{t_1}, t_1), \mathcal{E}_{\theta_2}(\mathbf{x}_{t_2}, t_2)\big)$

    $\theta \leftarrow \theta + \eta \nabla_{\theta_1} \mathcal{L}_{\text{COT}}(\theta_1, \theta_2)$

**until** convergence

---

## A.2 DATASETS

We use the following publicly available datasets as the sources $\mathbf{x}$ or targets $\mathbf{y}$: Amazon handbags and shoes Yu & Grauman (2014) to perform handbag→shoes (64×64); CelebA faces Liu et al. (2015) to perform male→female (64×64); outdoor images of MIT places database Zhou et al. (2014) and LSUN church dataset Yu et al. (2015) to perform outdoor→church (128×128); auto-detected edges on the M&Ms dataset Campello et al. (2021) and ACDC dataset Bernard et al. (2018) to perform edges→cMRI (128×128). All the coupled datasets are unpaired and randomly sampled during training.

For the proposed zero-shot image editing scenarios, we utilize the trained models on the aforementioned tasks, where no additional dataset is needed.

## A.3 HYPER-PARAMETERS

Despite the differences between our method and diffusion-based models, we use the recommendations in Karras et al. (2022) for the common hyper-parameters such as learning rate and number of discrete time steps ($N = 40$). We use the noise scale $\sigma = 1$ for all the tasks.

## A.4 TRAINING DETAILS

For the network structure and the training details of the neural OT models, we follow the recommendations of Korotin et al. (2022a). The neural OT models converge in 1-2 days on a single NVidia A40 GPU (48GB). The batch size during training is 64 for all the tasks.

For the encoder models $\mathcal{E}_\theta$, the network structure uses the recommendations in Song et al. (2023), and the models converge in 3-4 days on 4×NVidia A40 GPUs (48GB). The batch size during training is 128 for all the tasks.

## A.5 ABLATION STUDY DETAILS

We provide three alternatives as a comparison to ablate our training and/or sampling choices.

In particular, we first train the models using adjacent positive pairs $\langle \mathbf{x}_{t_k}, \mathbf{x}_{t_k+1} \rangle$ instead of the COT Pairs $\langle \mathbf{x}_{t_1}, \mathbf{x}_{t_2} \rangle$ provided by Eq.13. This alternative evaluates the importance of the chosen COT Pair formulation and emphasizes the connection between consistency models and contrastive learning.

Secondly, we choose an opposite direction to train the neural OT models in each task. For example, in the handbag→shoes task, instead of training a neural OT model $T(\mathbf{x})$ from the handbag dataset to the shoes dataset, we train a reverse neural OT model $T'(\mathbf{y})$ from shoes data $\mathbf{y}$ to handbag data $\mathbf{x}$. This alternative evaluates the paper's choice of the neural OT model's direction and verifies the formulation of COT Pairs.

Finally, we provide an optional sampling strategy to prove the effectiveness of our self-augmentation sampling strategy in COT Editor. After training the models, we implement an ancestral-like sampling strategy to generate the results:

$$\tilde{\mathbf{x}}_{t_k}^{(k)} = \frac{t_k}{t_{k-1}}\tilde{\mathbf{x}}_{t_{k-1}}^{(k-1)} + (1 - \frac{t_k}{t_{k-1}})\tilde{\mathbf{y}}^{(k)} + t_k(1 - t_k)\sigma^2\mathbf{z}_k \tag{20}$$

$$\tilde{\mathbf{y}}^{(k+1)} = \mathcal{E}_\theta\big(\tilde{\mathbf{x}}_{t_k}^{(k)}, t_k\big), \qquad k = 1, 2, \dots, \qquad \tilde{\mathbf{x}}_{t_0}^{(0)} = \mathbf{x} \tag{21}$$

## B  PROOF OF THEOREM

**Proposition 3.1.** *Let $\pi^*$ be the OT plan between $\mu(\mathbf{x})$ and $\nu(\mathbf{y})$. Let the OT map $T^*$ recover $\pi^*$. The augmentation defined by Eq.12 using $T^*$ samples the same probability as the dynamic extension of the EOT plan $\pi_\lambda^*$ with $\lambda = 2\sigma^2$.*

*Proof.* According to Gushchin et al. (2024), the augmentation between $\mathbf{x}$ and $T^*(\mathbf{x})$ using Eq.12 samples a probability distribution:

$$p_t(\mathbf{x}_t|\mathbf{x}, T^*(\mathbf{x})) = \mathcal{N}(\mathbf{x}_t|tT^*(\mathbf{x}) + (1 - t)\mathbf{x}, t(1 - t)\sigma\mathbf{I}) \tag{22}$$

which is the time marginal of a Brownian Bridge $\mathbf{w}_{|\mathbf{x},T^*(\mathbf{x})}^\sigma$ (Appendix C). Using the probability distribution in 22, the Schrödinger Bridge $S^*$ (Appendix D) between $\mu(\mathbf{x})$ and $\nu(\mathbf{y})$ can be estimated by:

$$\tilde{S}^* = \int_{\mathbb{R}\times\mathbb{R}} \mathbf{w}_{|\mathbf{x},\mathbf{y}}^\sigma d\tilde{\pi}^*(\mathbf{x}, T_\phi(\mathbf{x})) \tag{23}$$

Which is the dynamic extension of the entropy-regularized OT problem with optimum $\pi_{2\sigma^2}^*$ according to Tong et al. (2023), where the joint marginal distribution $\pi^{S^*}$ of $S^*$ at times 0,1 is the EOT plan $\pi_{2\sigma^2}^*$ in 7, i.e., $\pi^{S^*} = \pi_{2\sigma^2}^*$. $\qquad\square$

## C  BROWNIAN BRIDGE

Suppose we have a data point $\mathbf{x}$ with time intermediates $\mathbf{x}_t$ in the processes. Given a Wiener process $\mathbf{w}_t^\sigma$ defined by $d\mathbf{w}_t^\sigma = \sqrt{\sigma}d\mathbf{w}_t$ with volatility $\sigma > 0$, $t \in [0, T]$, and standard Wiener process $\mathbf{w}_t$. A Brownian Bridge is the conditional probability distribution $\mathbf{w}_{|\mathbf{x}_0,\mathbf{x}_T}^\sigma$ subject to the condition that the start and end point of the process is $\mathbf{x}_0, \mathbf{x}_T$. The probability distribution is:

$$\mathcal{N}(\mathbf{x}_t|t\mathbf{x}_T + (T - t)\mathbf{x}_0, t(T - t)\sigma\mathbf{I}) \tag{24}$$

Intuitively, the Brownian Bridge is pinned to the values $\mathbf{x}_0, \mathbf{x}_T$ at $t = 0$ and $t = T$, and the most uncertainty lies in the middle of the bridge.

## D  SCHRÖDINGER BRIDGE

Given two probability distribution $\mu(\mathbf{x})$ and $\nu(\mathbf{y})$, consider the Wiener process $\mathbf{w}_t^\sigma$ with volatility $\sigma > 0$ starts at $\mu(\mathbf{x})$ at $t = 0$, the Schrödinger Bridge between $\mu(\mathbf{x}), \nu(\mathbf{y})$ is:

$$S^* = \min_{S \in \mathcal{F}(\mu,\nu)} \mathrm{KL}(S \parallel \mathbf{w}_t^\sigma) \tag{25}$$

where $S$ is a stochastic process and $\mathcal{F}(\mu, \nu)$ is a set of stochastic processes with the start of $\mu(\mathbf{x})$ at $t = 0$ and end of $\nu(\mathbf{y})$ at $t = T$.

## E  DIFFUSION MODELS

Diffusion models learn to denoise the data in different noise scales and generate samples from noise via an iterative denoising process. The original data distribution $\mu(\mathbf{x})$ is diffused with a stochastic differential equation (SDE):

$$d\mathbf{x}_t = \mathbf{g}(\mathbf{x}_t, t)dt + \sigma(t)d\mathbf{w}_t \tag{26}$$

where $t \in [0, T]$, $T > 0$ is a constant, $\mathbf{g}$ is the drift term and $d\mathbf{w}_t$ represents a standard Wiener process. We denote the intermediate distribution of $\mathbf{x}_t$ as $p_t(\mathbf{x})$. Then the SDE process has a dual ODE whose solution trajectories at time $t$ are distributed according to $p_t(\mathbf{x})$:

$$d\mathbf{x}_t = \left[ \mathbf{g}(\mathbf{x}_t, t) - \frac{1}{2}\sigma(t)^2 \nabla \log p_t(\mathbf{x}_t) \right] dt \tag{27}$$

where $\nabla \log p_t(\mathbf{x}_t)$ denotes the score function of $p_t(\mathbf{x})$, which is estimated by a neural network $\mathbf{s}_\phi(\mathbf{x}_t, t) \approx \nabla \log p_t(\mathbf{x}_t)$. We then sample $\mathbf{x}_0$ from the estimated probability flow ODE:

$$\frac{d\mathbf{x}_t}{dt} = -t\mathbf{s}_\phi(\mathbf{x}_t, t) \tag{28}$$

where we initialize $\mathbf{x}_T \sim \mathcal{N}(\mathbf{0}, T^2\mathbf{I})$ and solve 28 backward in time to obtain the generation $\hat{\mathbf{x}}_0$ via various ODE solvers such as Euler and Heun solvers.

## F  LIMITATIONS

COT Flow explicitly builds the bridge between optimal transport and diffusion/flow-based models. However, our method requires a two-step training pipeline, including the neural OT model $T(\mathbf{x})$ and the encoder model $\mathcal{E}$, which may influence the training and deploying stability. A promising future direction is to design an end-to-end method with OT formulation explicitly.

## G  BROADER IMPACTS

COT Flow and other generative models pose a risk of synthesizing inappropriate content such as deep-fake images, violence, or privacy-related offensiveness.

## H  ADDITIONAL EXPERIMENTS

We compared the zero-shot image editing ability between our model and SDEdit on different datasets. We took 600k iteration steps with a batch size of 256 on 4×NVidia A40 GPUs to train our model, and we followed the recommended training hyper-parameters in Meng et al. (2021) to ensure the convergence of the baseline SDEdit model. The results in Fig.6 show that our model (COT Editor) outperforms SDEdit in both image editing quality and fidelity on all datasets.

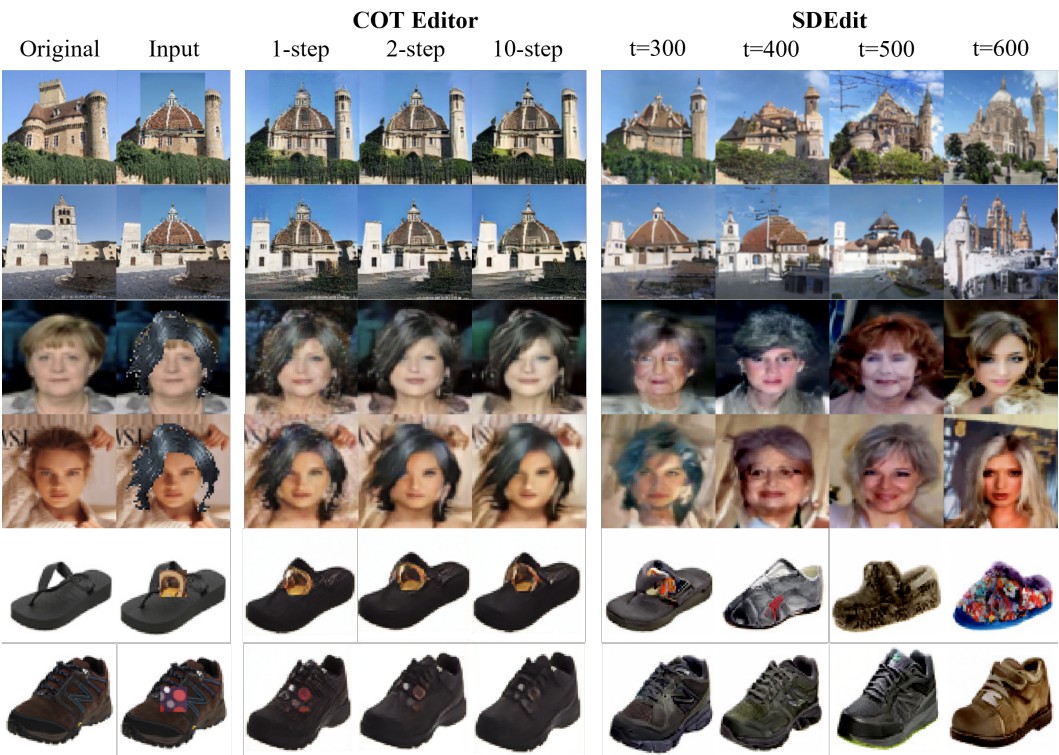

Figure 6: Zero-shot image editing comparison between our method (COT Editor) and SDEdit on CelebA male→female (64×64), handbag→shoes (64×64), and outdoor→church (128×128). We use one-step and multi-step sampling in our method and set $t = 300, 400, 500, 600$ of the reverse diffusion process in SDEdit to perform the editing results.

