# OpenReview forum: "COT Flow: Learning Optimal-Transport Image Sampling and Editing by Contrastive Pairs"
_ICLR.cc/2025/Conference — Submitted to ICLR 2025_

### Official Review · Reviewer_v1MD · 2024-10-28

**Soundness:** 3
**Presentation:** 2
**Contribution:** 3
**Rating:** 5
**Confidence:** 4

**Summary:**

This work presents Contrastive Optimal Transport Flow (COT Flow), a new method that achieves fast and high-quality generation with improved zero-shot editing flexibility compared to previous diffusion models.

**Strengths:**

This work presents Contrastive Optimal Transport Flow (COT Flow), a new method that achieves fast and high-quality generation with improved zero-shot editing flexibility compared to previous diffusion models.

**Weaknesses:**

1. Experiments are not enough only on handbag→shoes (64×64), CelebA male→female (64×64), and outdoor→church (128×128). These tasks are unuseful.
2. The images are too small and unclear.
3. The comparison methods are too old. It should compare with at least some of the latest text-based image editing methods in 2024.

**Questions:**

1. How to use this flow to open-set text-based image editing? What is the source distribution and target distribution?
2. What is the role of \phi_\omiga in Eq.10 and 11? Eq.10 and 11 confuse me.
3. In Algorithm 1 COT Training, if the encoder learns to output all zero, the loss function will be zero, how to handle this problem?

I will be happy to raise the rating if the response is good.

---

### Official Review · Reviewer_pPKV · 2024-11-01

**Soundness:** 2
**Presentation:** 2
**Contribution:** 2
**Rating:** 3
**Confidence:** 4

**Summary:**

This paper proposes the COT Flow model by integrating neural optimal transport and consistency models. Based on the similarities between contrastive learning and consistency models, the authors introduce a new method for defining positive pairs. Furthermore, they demonstrate that the COT Flow can be applied to zero-shot image editing.

**Strengths:**

1. The proposed COT Flow model effectively enables unpaired image-to-image translation.
2. The COT Flow model shows potential for various zero-shot image editing scenarios.

**Weaknesses:**

1. Comparison methods are limited.
2. Quantitative evaluations are limited, relying solely on FID scores.
3. Qualitative results lack visual impact.

**Questions:**

1. Does COT Flow truly address the generative learning trilemma? The comparison only includes FID scores, but if the model claims to tackle the trilemma, it should demonstrate comparisons in terms of sampling speed, quality, and diversity against existing methods.
2. What is the training process for the neural optimal transport model?
3. Comparison methods are limited. Numerous recent studies on diffusion-based unpaired image-to-image translation tasks should be included, such as [1], [2], and [3]. Additional comparisons with more recent or relevant baselines could strengthen the validity and impact of the findings.

[1] Korotin, A., Selikhanovych, D., & Burnaev, E. Neural Optimal Transport. In The Eleventh International Conference on Learning Representations.
[2] Su, X., Song, J., Meng, C., & Ermon, S. Dual Diffusion Implicit Bridges for Image-to-Image Translation. In The Eleventh International Conference on Learning Representations.
[3] Zhao, M., Bao, F., Li, C., & Zhu, J. (2022). Egsde: Unpaired image-to-image translation via energy-guided stochastic differential equations. Advances in Neural Information Processing Systems, 35, 3609-3623.

---

### Official Review · Reviewer_eUMd · 2024-11-04

**Soundness:** 2
**Presentation:** 2
**Contribution:** 2
**Rating:** 5
**Confidence:** 3

**Summary:**

This paper present Contrastive Optimal Transport Flow (COT Flow), a method that achieves fast and high-quality generation with improved zero-shot editing flexibility compared to previous diffusion models.

**Strengths:**

The paper is well-written

**Weaknesses:**

1.The performance improvement of the paper is not significant.

2.The comparison method is outdated; SDedit is a work from two years ago.

This paper has neither impressive results nor significant improvements. I did not find any highlights in this paper, leaning towards a rejection.

**Questions:**

see above

---

### Meta-Review · Area_Chair_HhS8 · 2024-12-12

**Metareview:**

The paper introduces the Contrastive Optimal Transport Flow (COT Flow), aimed at improving the speed and quality of image generation, with a focus on zero-shot editing flexibility. The model integrates neural optimal transport with consistency models for defining positive pairs in contrastive learning.

***Strength:***
- The paper is clearly written and presents a novel integration of neural optimal transport with consistency models.
- Demonstrates potential for application in zero-shot image editing scenarios.

***Weaknesses:***
- The performance improvement over existing methods is marginal.
- Uses outdated comparison methods and lacks significant comparative analysis with recent state-of-the-art models.
- Limited scope and depth in experiments which only on small-scale tasks with low-resolution images.
- Both qualitative and quantitative evaluations are weak, failing to provide compelling evidence of the model's effectiveness.

The authors did not submit a rebuttal to address the concerns raised during the review process, missing the opportunity to clarify the highlighted issues. Given all the concerns remaining, this paper cannot be accpeted at this time.

**Additional Comments On Reviewer Discussion:**

There is no rebuttal from the author. All concerns persist and all reviewers gave negative scores.

---

### Decision · Program_Chairs · 2025-01-22

Reject